# Emotional Labor in Pediatric Palliative Care: A Scoping Review

**DOI:** 10.3390/nursrep15040118

**Published:** 2025-03-26

**Authors:** Ana Inês Lourenço da Costa, Luísa Barros, Paula Diogo

**Affiliations:** 1Nursing Research Innovation and Development Centre of Lisbon (CIDNUR), Nursing School of Lisbon (ESEL), 1600-190 Lisbon, Portugal; pmdiogo@esel.pt; 2Child and Youth Nursing Department, Nursing School of Lisbon (ESEL), 1649-013 Lisbon, Portugal; 3Research Center for Psychological Science CICPSI, Faculdade de Psicologia, Universidade Lisboa, 1649-013 Lisbon, Portugal; lbarros@psicologia.ulisboa.pt

**Keywords:** emotions, palliative care, children, healthcare professionals

## Abstract

**Background:** Caring for children in palliative care especially impacts healthcare professionals’ personal and professional lives. Their emotional experience and needs are frequently forgotten. Healthcare professionals face emotional demands when caring for children with palliative needs and their parents. **Objective:** This scoping review aims to identify and map the scientific production about the emotional labor of healthcare professionals in pediatric palliative care. **Methods:** This scoping review was conducted according to the JBI recommendations and the PRISMA Extension for Scoping Reviews. We searched 16 electronic databases in August 2023 and updated the search on 17 February 2025. Articles were screened according to eligibility criteria, and a content analysis allowed for a summary of key findings. **Results:** Eleven publications were selected. Most studies were conducted in the United States of America and with nurses as the professionals involved. Many publications were qualitative studies and developed in a neonatal intensive care context. Using content analysis, five themes were identified: (1) emotional experience of healthcare professionals, (2) relational context involved, (3) managing professional and personal boundaries, (4) intrapersonal strategies of emotional labor, and (5) social and organizational strategies of emotional labor. **Conclusions:** The importance of implementing emotional labor strategies is highlighted, especially intrapersonal, social, and organizational strategies. Education, training, and reflection are needed within a workplace culture that recognizes emotional experiences and supports the emotional management of healthcare professionals. Emotional labor in pediatric palliative care should be recognized. Further research in this area is needed.

## 1. Introduction

More than 21 million children with life-threatening and life-limiting conditions will benefit annually from a palliative care approach, and more than eight million need specialized pediatric palliative care worldwide [1]. The World Health Organization [2] defines palliative care for children as the active, total, and holistic care of the child and family, starting when the disease is diagnosed and continuing during the treatment. Healthcare professionals aim to alleviate children’s physical, psychological, and social distress through an interdisciplinary approach involving the family and the available community resources. Care should be developmentally adequate and in accordance with family values. Despite variations in the definition of children’s palliative care [3,4], the concept and philosophy of care are consistent with care provided for children with life-threatening and life-limiting conditions and their families.

When providing palliative care to children and their families, healthcare professionals are exposed to various intense emotional experiences related to diagnoses, disease progression, death and dying, and grief. Sometimes, they manage these emotions using multiple strategies, but it is challenging to limit the expression of strong emotions [5]. The need to address these emotions makes pediatric palliative care an emotional labor-inducing experience [6].

Hochschild [7] studied the concept of emotional labor and proposed a theoretical framework that can be useful in understanding healthcare professionals’ necessary work to balance the needs of self and others and the demands of healthcare organizations [7]. According to this author [7], emotional labor involves the emotional processes occurring in care interactions. It implies that “the induction or suppression of feeling to sustain an outward appearance that produces in others a sense of being cared for in a convivial safe place” (p. 7). The concept of emotional labor is central to professions requiring face-to-face or voice-to-voice contact with the public, requiring the worker to produce an emotional state in another person, and allowing the employers to have some degree of control over the workers’ emotional activities through training and supervision [7]. Managing emotions effectively is one way for employees to achieve organizational goals: to achieve employer expectations, including surface acting and deep acting [7]. Another element of emotional exchange is the feeling rules, which are patterns utilized in emotional conversation to achieve what is correct. Therefore, feeling rules represent the emotions people should express as stated by social roles [7]. Hochschild [7] pointed out that emotional labor embraces the public and private domain. So, the private domain includes a person’s home, family, and friends. In contrast, the public domain is related to the work context. In this public domain, emotional labor is linked to a monetary exchange or commercial gain.

Surface acting occurs when one changes the outward expression to display expected emotions without changing the actual feeling [7]. In deep acting, people attempt to change their feelings to produce a more genuine emotional display in line with the emotions expected [7]. These strategies can be used together in the same interaction [8]. While surface acting implies a superficial display of feelings that may cause emotional dissonance, deep acting affects personal feelings and requires knowledge of emotion regulation processes [7]. Surface acting strategies can cause high personal and relational costs. On the other hand, deep acting leads to positive consequences like well-being, despite personal efforts [8].

The emotional labor of healthcare professionals is particularly needed when they work in distressing situations [9]. It involves a collaborative and relational process to develop a therapeutic relationship during healthcare professionals’ involvement with clients [10,11]. Healthcare professionals, specifically nurses, must use complex emotional reflexivity to recognize and address their emotional needs [8,12]. However, emotional labor is “often perceived as an undervalued” [10] (p. 2) and under-appreciated aspect of the care work [13] or an under-reported, invisible component of the service [9], identified as a “tacit and uncodified skill” [14] (p. 253). Emotional labor involves emotional regulation strategies available to nurses and nurses managing their own and their clients’ emotions [10,15]. Working in pediatric palliative care services especially impacts healthcare professionals’ personal and professional lives. Their emotional experience and needs are frequently forgotten. The grieving of healthcare professionals remains unknown because society expects them to be strong in the face of children’s death [16].

A preliminary search was conducted in SCOPUS, Web of Science, and JBI Evidence Synthesis to identify similar scoping reviews, and no current or underway systematic reviews or scoping reviews on the topic were identified. So, research on healthcare professionals’ emotional labor in pediatric palliative care is scarce. In 2020, a scoping review aimed to identify and systematize the availability of publications about emotional labor in diverse pediatric nursing care contexts and demonstrated that the most studied pediatric nursing contexts were pediatric inpatient services, pediatric palliative care, and neonatal intensive care [17]. A reflective study based on theoretical and experiential aspects of emotional care in pediatric nursing highlighted professional competence in emotional care for children and parents [18]. So, identifying the available evidence regarding emotional labor in pediatric palliative care is relevant to understanding specialized competencies for managing emotions in a particular context of pediatric palliative care. Considering this gap, we aimed to answer the following review question: how do healthcare professionals perform emotional labor when caring for children and their parents in palliative care? This scoping review aims to identify and map the scientific production about the emotional labor of healthcare professionals in pediatric palliative care.

## 2. Materials and Methods

This scoping review was conducted according to the Joanna Briggs Institute’s methodology for scoping reviews [19] through the following steps: eligibility criteria, search strategy, sources of evidence selection, data extraction, and data analysis and presentation. The recommendations defined in the Preferred Reporting Items for Systematic Reviews and Meta-Analyses Extension for Scoping Reviews (PRISMA-ScR) Checklist [20] (Appendix A) were followed. The research protocol was registered in the Open Science Framework Registries (https://doi.org/10.17605/OSF.IO/2T6G8 (accessed on 15 March 2025)).

### 2.1. Eligibility Criteria

The eligibility criteria were based on the PCC framework (participants, concept, and context) recommended by Joanna Briggs Institute [21], the participants were healthcare professionals, the concept was emotional labor, and the context was pediatric palliative care. Thus, the review question was formulated through the PCC framework: how do healthcare professionals perform emotional labor when caring for children and their parents in palliative care?

The eligibility criteria were: (1) publications referring healthcare professionals who care for children and their parents with palliative needs (nurses, physicians, psychologists, social workers, pediatric palliative care team); (2) publications including the concept of emotional labor or emotion work, or the related concepts of emotional management or emotional regulation; (3) peer-reviewed articles, opinion articles, editorials, Master’s dissertations, doctoral theses, guidelines, commentaries, technical reports, reviews, and policy briefs; (4) no time limit; (5) publications written in English and Portuguese; and (6) publications with the full-text available (authors were contacted when the full text was not immediately available). We excluded all publications referring to healthcare students (when they were the only participants) and healthcare professionals occupying management positions.

### 2.2. Search Strategy

A computerized literature search was conducted in August 2023 and updated on 17 February 2025. A total of 16 electronic databases were used, including Web of Science, SCOPUS, CINAHL Complete, MEDLINE Complete, Psychology and Behavioral Science Collection, JBI OVID, Scielo, Open Access Theses and Dissertations (OATD), Google Scholar, PubMed, Lilacs, ScienceDirect, ResearchGate, Biomed Central, British Library Collection, and Open Access Scientific Repository of Portugal (RCAAP).

Considering this is a scoping review aiming to determine the coverage of literature on the topic of emotional labor in the context of pediatric palliative care, we decided to include Master’s dissertations and doctoral theses. To capture these studies, we included databases including these publications, such as the British Library Collection, Open Access Theses and Dissertations (OATD), and Open Access Scientific Repository of Portugal (RCAAP). We also selected Scielo and Lilacs databases to capture studies in Portuguese in scientific journals. JBI OVID was selected to include scoping reviews and systematic reviews, and Google Scholar and ResearchGate to identify other full-texts and grey literature.

First, an initial search of MEDLINE Complete was undertaken to identify articles on the topic. The search strategy used in MEDLINE Complete was (Nurses OR Physicians OR Social Workers OR Psychologists OR Patient Care Team) AND (Emotion Work OR Emotional Management OR Emotional Regulation OR Emotional Labour OR Emotional Labor) AND (Pediatric Palliative Care OR Paediatric Palliative Care OR End-of-life). These descriptors were combined using the Boolean operators OR and AND and were used in databases for information retrieval (Table 1).

### 2.3. Sources of Evidence Selection

Following the search, the Rayyan QCRI platform was used to facilitate the archiving, organization, and selection of articles. Two independent authors (A.I.L.C., P.D.) screened the titles and abstracts of publications and chose the studies for full reading. Reasons for excluding publications in the full text that did not meet the inclusion criteria were recorded. Two independent authors (A.I.L.C., P.D.) undertook the final study selection. The third author (L.B.) analyzed disagreements to reach a consensus.

The search results and the study inclusion process were presented in a PRISMA flow diagram.

### 2.4. Data Extraction

Two independent reviewers used a data extraction tool developed by the authors to extract data from publications included in the scoping review. The information was organized in a table, including author(s), publication year, country, publication title, methodology, participants, aim(s), main results, and limitations/suggestions. Any disagreement was resolved through discussion with the third reviewer, who played the role of judge in the previous phase. Two authors (A.I.L.C., P.D.) extracted the data from the source of evidence to answer the review question, and a third author (L.B.) checked the data extracted and helped resolve disagreements.

### 2.5. Data Analysis and Presentation 

The authors used content analysis to synthesize and summarize the results of the scientific production and characterize the emotional labor in pediatric palliative care.

Potential limitations of the synthesis process related to variability among the studies identified were discussed between authors and managed in a manner that a summary of results corresponds with the review question and objective.

## 3. Results

### 3.1. Characteristics of the Publications

PRISMA (Figure 1) shows the selection and screening process flowchart. We found eleven studies that met the inclusion criteria (Table 2). Publications spread across 16 years (from 2008 to 2024), with a higher number occurring between 2021 and 2024. Most publications were from the United Kingdom and the United States of America. Many were qualitative studies with different methodological approaches, and only one used a quantitative methodology. We found one systematic review and two narrative reviews. The population studied was mostly nurses, with a few studies with physicians and other healthcare professionals. Studies were conducted in five different contexts, with the neonatal intensive care unit the most frequent. Some studies took place in more than one context, and several did not clarify the context of care (Table 3).

### 3.2. Characterization of Healthcare Professionals’ Emotional Labor

Five themes characterizing the emotional labor of healthcare professionals in pediatric palliative care emerged from the selected publications: healthcare professionals’ emotional experience, relational context involved, managing professional and personal boundaries, intrapersonal emotional labor strategies, and social and organizational emotional labor strategies. Providing care to children with palliative needs and their parents involves building trust and a close relationship between healthcare professionals, children, and parents over time. This relational context caused intense and negative emotions in healthcare professionals, who needed to manage their professional and personal boundaries. As a result, healthcare professionals regulated their intense emotions by implementing intrapersonal, social, and organizational emotional labor strategies (Figure 2).

#### 3.2.1. Healthcare Professionals’ Emotional Experience

Healthcare professionals dealt with complex and traumatic situations when caring for children with palliative needs and their parents. In palliative care, nurses could experience burnout, grief, and stress because of continuously hiding their emotions [32]. They felt negative emotions, such as sadness, dismay, and anxiety, especially in caring for children and parents for long periods. Additionally, nurses felt helpless, mainly when the child eventually dies, and this experience affects their lives [26]. Nurses, doctors, and other healthcare professionals felt inadequate and powerless in specific situations, such as when nothing they could do would have made a difference or thought they had not done enough to help children and parents [27]. The decision to end the cardiopulmonary resuscitation is felt like a death sentence [29]. The tragedy of the loss and the parents’ reaction emotionally overwhelm healthcare professionals [29]. Those who cared for children with life-limited conditions and their parents are continuously exposed to the child and family grief.

Children and their parents felt frustration and anger that they transmitted to nurses [32]. Nurses deal with their own emotions and facilitate the emotional management process of pediatric patients [32]. So, the emotions experienced by nurses and parents are often described as contagious [31]. The intense emotions and grief of parents who were upset, scared, or angry were caused by situations of unfairness and the unnatural death of a child, resulting in suffering for healthcare professionals who witnessed them [27] (Table 4). Healthcare professionals who were parents faced different challenges because of their personal parenting experiences and their work of caring for children with palliative needs.

Nurses accompanied the child and parents in good and bad times. They were exposed to different and intense emotions [33] due to the uncertainty in outcomes for clinically unstable children and the dying process. Nurses deal with intense emotions when caring for children with complex diseases and their parents, using their competence and professionalism to promote quality of care [31].

#### 3.2.2. Relational Context Involved

The context involved in providing care to pediatric clients showed how the emotional dimension of care was intrinsic to the interactive and relational process, with an intense emotional exchange between healthcare professionals, children, and parents. In most situations, the healthcare professionals developed close, supportive relationships with the children and parents [31], although interactions may be difficult because of the amount of stress. Managing relationships in caring for children with palliative needs and parents involved a purposeful positioning in which intentionality, intimacy, and one-to-one interaction [23,25], not usually experienced outside family relationships, are described [33]. During the care process, intense human relationships were established that led nurses to be responsive to verbal and nonverbal signs of the child’s and parents’ behavior [33].

Physicians highlighted the importance of building a planned and strategic relationship between physicians, children, and parents, where trust was fundamental, especially if they needed to accompany families through the dying process, showing its utility in future discussions about end of life [24]. This relationship, established since the first meeting, was essential in providing emotional support and helping physicians, children, and parents to discuss medical and nursing interventions that might reduce suffering at the end of children’s lives [24].

Physicians stated that they intentionally cultivate relationships to achieve their goals of helping families through direct medical aspects of a child’s care and encouraging parents and children to engage in end-of-life planning [24]. In addition, nurses allowed parents to vent their concerns without feeling personally attacked [31]. Offering empathy and compassion and meeting family needs were challenging for healthcare professionals who sought to provide families with the proper emotional support and comfort [27]. Therefore, nurses involved the parents in non-threatening discussions about movies, restaurants, and social activities to provide a mental break and distraction and facilitate the maintenance of relationships [31]. Occasionally nurses shared superficial personal information with parents intending to make a connection and build trust. However, the level of information shared between nurses and parents varies according to the family, circumstances, and duration of the relationship [23].

A close relationship between children, parents, and nurses led to an appropriate environment to facilitate accepting parental choices, the operational aspects that promote death at home, monitoring the family, and the reorganization of services. Furthermore, this relationship supported pediatric clients in different ways: information, guidance, emotional, instrumental, and financial [30]. Multiple characteristics of the relationships between nurses, children, and parents were described, such as active interest, affection, kindness, flexibility, responsibility, sensitivity, listening, and open communication [30], which allowed comfort, pain relief, and child-centered care [30]. Intrinsic to the relationship between healthcare professionals, children, and parents, some authors [27,30] valued effective communication, which involved the challenge of successfully answering children’s and parents’ questions, being honest about the child’s prognosis, having discussions around end-of-life care goals and needs, making sure to involve the family in these decisions, respecting their wishes, and helping the expression of feelings and thoughts (Table 5).

#### 3.2.3. Managing Professional and Personal Boundaries

In pediatric palliative care, the healthcare professionals’ relationship with the child and family was maintained for an extended period, so it was difficult not to create an attachment to the child and parents and some level of proximity and intimacy. However, healthcare professionals must maintain professional boundaries while providing family-centered care [33]. Brimble’s study [25], which aimed to understand how nurses maintain professional integrity while providing long-term practical, emotional, social, and spiritual care to children and parents, demonstrated that nurses used strategies to manage relationships with parents based on interactions precisely balancing personal life and professionalism. Nurses search for a balance between being sociable and personable with families and simultaneously being professional [23].

Nurses, doctors, and other healthcare professionals managed their emotions and emotional responses, and it was essential to acknowledge the need for professional and personal boundaries, where self-care was vital in building resilience and maintaining strength in the face of children’s death [27]. Physicians needed to be aware of balancing emotional labor and personal well-being, reinforcing that it was critical to maintain the emotional reserves they needed to help the families they accompany. Maintaining some form of personal life separate from their work was the key to managing intense emotions resulting from their work [24].

Gengler [24] highlighted the tension between the commitment to emotionally supporting children and parents and the emotional care for themselves, which is particularly significant in physicians’ decisions about funeral or memorial services attendance and how much was available outside formal channels [24]. Healthcare professionals attended patients’ funerals to demonstrate the extent to which providing emotional support to families was central to their identities [24,31]. They described this as a great honor and an opportunity to share what they admired about children they had cared for [24,31] (Table 6). However, Chinese nurses did not talk much about the feelings of children dying and were not invited to attend the children’s funerals [26].

#### 3.2.4. Intrapersonal Strategies of Emotional Labor

Healthcare professionals face challenges associated with regulating their more intense emotions, including avoiding becoming overwhelmed or letting their sadness hinder their ability to provide adequate care [27]. Several publications of this revision that specifically refer to emotional labor strategies used by healthcare professionals when caring for children and their parents focused on how healthcare professionals manage their own emotions in pediatric palliative care. Erel and Büyük [28] observed that nurses tended to suppress their real emotions and behave superficially while providing neonatal palliative care. Through surface acting, nurses put on an appropriate face when dealing with difficult and traumatic situations in caring for dying children and their families [32]. Healthcare professionals did not consider this outward expression of emotions acceptable in caring for neonates and parents [29]. Brimble et al. [23] underlined that nurses developed many activities based on having fun with children and parents in hospice, and in these situations, nurses suppressed distress and induced joviality.

The dying process was particularly challenging for healthcare professionals due to the dimensions of actual or potential losses and the fear of one’s finiteness; this might cause them to keep emotional distance as a protective mechanism [30]. Uhrecký et al. [29] claimed that distancing from affective responses prevented healthcare professionals from being overwhelmed and maintaining professionalism, which was inherent to their role. In the same circumstances, healthcare professionals built an emotional barrier to avoid becoming emotionally attached to the children and parents [33] and maintained a sense of separation in their lives. Indifference was also a defense mechanism of protection from the process of finiteness [30].

Task orientation was a strategy for nurses to focus on the specific care of children, keeping work sustainable, deal with uncertainty, and attain the affective neutrality expected from nurses [33]. Likewise, the Uhrecký et al. [29] study, which explored the healthcare professionals’ emotion regulation strategies in a simulated task, demonstrated that healthcare professionals focused on the task of caring for children based on attentional narrowing, concentration in a caring plan, being vigilant for potential sources of danger, and distancing from the caring scene because they saw emotions as distractors. Similarly, to reduce the emotional pain caused by caring for children with complex conditions and their parents, nurses reduced interactions with parents, focusing on the tasks of caring for neonates, procedures, and monitoring, using super-efficient attention to detail when caring for babies with complex diseases [31]. However, the uncertainty inherent to this situation caused a perception of a lack of control and stress [31].

Sometimes, the intense stress of these situations might lead to nurses asking for a service transference, describing how painful it was to witness the children’s crying and the parents’ distress [31].

In the Uhrecký et al. [28] study, healthcare professionals used cognitive reinterpretation to help them emotionally accept the death of a child by restructuring this event as a positive output for a severe medical condition, attributing the child’s death to external factors and parents’ reactions to their child’s loss. This strategy was important to deal with personal guilt and continue to care for children and parents. Similarly, Cricco-Lizza’s [31] findings revealed the importance of taking breaks during nurses’ working days to cope with the pressure of the neonatal intensive palliative care environment. Despite the uncertainty of results, the nurses continued caring for neonates and parents, finding meaning in their everyday work. Moreover, making memories with and for parents seemed to be a satisfying activity with emotional and personal benefits [23].

Furthermore, physicians who had young children of their own need to “compartmentalize” the fear that their children might become ill or die while surrounded by sick and sometimes dying children [24] (p. 7). In this sense, Rawlings et al. [27] and Cricco-Lizza [31] explained that nurses, physicians, and other healthcare professionals who are parents think of their children and what would happen if they lost them.

Attending memorial services and funerals was a formal mechanism that provided a socially acceptable mourning ritual and gesture of family support [31]. These contributed to strengthening relationships between physicians and parents in numerous end-of-life contexts while being sensitive to the emotional demands involved in work [24].

Regarding self-care, Cricco-Lizza [31] stated that nurses use individual strategies outside their professional role, such as exercise, rest, recreational activities, spiritual restoration, listening to music, and reading a book. Moreover, the study of Bian et al. [26] that explored pediatric nurses’ coping strategies in caring for dying children demonstrated the importance of some activities like self-meditation, watching TV, and slowly forgetting the experience of rescuing the children (Table 7). Diogo et al. [30] also referred to some personal characteristics like high optimism and self-esteem, the ability to attribute a positive meaning to the experience of negative emotions, spirituality, and religion as crucial to facing adversity.

#### 3.2.5. Social and Organizational Strategies of Emotional Labor

This theme related to how emotional labor was performed to support relationships with healthcare professionals inside a team and organizational hierarchy. In Bian et al. [26], nurses searched for friends or colleagues to discuss the intense experience of caring for children with palliative needs and parents. However, Cricco-Lizza [31] highlighted that some nurses did not reach out to their own families to share these experiences because they thought they would not understand the nature of their work and would be unable to support them.

Another strategy of emotional labor was the use of humor between team members. Humor revealed a high level of trust between the team members, the recognition of each member’s skills, and how those skills could support the management of emotions in the workplace [33]. The results of the Cricco-Lizza [31] study demonstrated that nurses use social talk and humor as a strategy to move away from the emotional intensity of care.

Maunder [33] highlighted the importance of nurses sharing experiences inside the team and becoming part of a recognized palliative care team. Nurses identified peer support and service delivery structure as vital for managing the emotional challenges of caring for children with palliative needs and their families [23]. Specifically, Cricco-Lizza [31] pointed out that nurses needed to search for support and security from their managers for distressing emotions. They actively intervened to promote a supportive work environment sustained by team spirit, kept the unit well-staffed, and provided the necessary resources. Diogo et al. [30] reinforced that nurses sought informational support inside the team. Bian et al. [26] stated the importance of receiving approval from the leadership, highlighting that nurses desired support from a leader, especially support and encouragement from the nurse manager. A simple gesture of approval could reduce the pressure of care. Inside the team, it was fundamental to clarify that a child’s death was not the responsibility of a nurse. Maunder [33], in her narrative review, argued that emotional labor continued to be undervalued by managers and healthcare services.

According to Erel and Büyük [28], neonatal intensive care nurses recognized the need for education and training programs incorporating the emotional labor concept and palliative care, promoting professional development. In Gengler’s [24] study, physicians recognized the importance of formal medical training to prepare them to care for children with palliative needs and parents. Cricco-Lizza’s [31] results demonstrated that nurses needed educational sessions to focus on critical care competencies, policies, procedures, and debriefing sessions (Table 8).

## 4. Discussion

This scoping review described essential features of the emotional labor of healthcare professionals in pediatric palliative care. Five central themes emerged: (1) healthcare professionals’ emotional experience, (2) the relational context involved, (3) managing professional and personal boundaries, (4) intrapersonal strategies of emotional labor, and (5) social and organizational strategies of emotional labor. These themes were not independent of one another, highlighting that emotional labor in pediatric palliative care is a complex and inevitable component of care [34].

Our results showed an increasing number of publications since 2021, probably explained by the growing relevance attributed to healthcare’s emotional dimension and emotional labor conceptualization’s development. Since Hochschild [7] conducted a study in sociology, various disciplines have explored the concept of emotional labor in healthcare professionals’ practice, including nursing [35]. Theodosius [11] affirmed that emotional labor is particularly relevant for healthcare and social care professions.

Most publications included in this review were qualitative studies aiming to describe the subjective experience of healthcare professionals who care for children with palliative needs and their parents and the aim to understand the global emotional experience [36]. Most studies focused on nurses’ experiences, emphasizing nurses’ emotional labor resulting from continued contact with children and parents and reinforcing emotional labor as an integral part of the nursing routine [15].

According to Smith [10], emotional labor is defined as the competencies involved in care and recognition of the emotions of patients and the emotions experienced by healthcare professionals, and this perspective of emotional labor was present in our results. These results focused on healthcare professionals’ emotional labor, highlighting the importance of the relationship established between healthcare professionals and pediatric clients, which causes the need to develop emotional management strategies. Therefore, the triple centrality of emotional labor proposed by the model of emotional labor in pediatric nursing [37] was implicit in our results because emotional labor was focused on healthcare professionals but occurred predominantly in a relational context with parents and children. The emotional labor model in pediatric nursing demonstrated the importance of nurses’ emotional regulation to be emotionally available to nurture and care with affection, build stable relationships, promote a safe and kind environment, and positively affect the management of the emotional state of the children and their parents [37].

However, to answer the needs of children and parents, the process of managing healthcare professionals’ emotions is complex because they manage intense personal emotions while at the same time responding to the pediatric client’s emotions.

As expected, research showed that pediatric palliative healthcare professionals used emotional labor in their clinical practice. It seems that healthcare professionals regulate their emotions beyond the expectation of their employer, in contrast with the Hochschild model [7], which refers to people managing emotions effectively to meet employer expectations in emotionally challenging situations. In our results, emotional management occurred in the relational process involved in caring for children and parents, in other words, by building a relationship. This idea was also evident in the study of Taylor, Smith, and Taylor [38], which explored the experience of health practitioners when working with families to safeguard children and the invisibility of the emotional work involved.

Managing professional and personal emotional boundaries when caring for children with palliative needs and their parents was related to the feeling rules about what emotions healthcare professionals should show and the degree of that expression consistent with social roles [7]. This management of emotions was so complex for healthcare professionals because private (home) and public (workplace) spheres coexist, making establishing and maintaining limits difficult [7]. According to Hochschild [7], emotion management occurs in the private sphere, while emotional labor is related to the workplace or frontstage.

Healthcare professionals felt stressed when confronting the suffering of clients in palliative care, so they needed to implement various emotional labor strategies to adapt to adverse circumstances. These strategies could potentially benefit healthcare professionals’ physical and mental health and reduce emotional suffering, resulting in the relationship between healthcare professionals and pediatric clients. This scoping review results underlined intrapersonal emotional labor strategies based on surface acting, in which healthcare professionals consciously use their bodies to change emotions or feelings to correspond to what they want to express [7], such as suppressing real emotions and behaving superficially, and task orientation. Additionally, healthcare professionals used emotional distance as an emotional regulatory strategy associated with surface acting. Hochschild [7] also considered emotional dissonance as a surface-acting strategy because people genuinely felt emotions different from the emotions displayed. When healthcare professionals use surface acting strategies, the outcomes can be dissatisfaction, stress, and burnout and can contribute to emotional exhaustion [39]. Grandey [40] suggested that surface acting is related to response-focused emotion regulation, in which healthcare professionals use suppression to change emotional expression and behavior after an emotion is felt. Gray [41] argued that nurses frequently must regulate their emotions to achieve patient satisfaction, and their own emotions, such as sadness, anger, and helplessness, tend to remain hidden and need to be explored.

In contrast, in deep acting, healthcare professionals change emotions or feelings from the inside, showing genuine emotions using a variety of strategies [7], such as verbal and physical stimuli [10]. Deep acting was a desirable strategy to adopt for healthcare professionals using different strategies because healthcare professionals could feel satisfaction, pride, and gratitude when taking care of children with palliative needs and their parents, applying imagination, reflection, training, and supervision. Results demonstrated that healthcare professionals utilized some intrapersonal emotional labor strategies related to deep acting as self-care and cognitive reinterpretation, even as the support of the team and leadership as a facilitator of social and organizational emotional labor strategies. Deep acting positively affected healthcare professionals by increasing their sense of personal accomplishment [42]. In deep acting, antecedent-focused emotion regulation was applied to change an event through reappraisal and distraction, such as social talk and humor.

Attending memorial services and funerals were included in two themes: managing professional and personal boundaries and intrapersonal emotional labor strategies. The Schoenbine et al. [43] study argued that healthcare professionals believed that attending funerals respected professional boundaries, and funeral attendance was associated with some norms as healthcare professionals, who care for children and parents for a long period. On the other hand, healthcare professionals maintaining contact with parents after the child’s death was crucial to evaluating family needs, giving bereavement support, and maintaining continuity of care. Attending funerals was a practice that helped parents feel more comfortable knowing that their child was remembered, facilitated meaning-making, and promoted a sense of closure [44].

Smith [10] highlighted that emotional labor should take part in the education about the dying process to prevent healthcare professionals from adopting distancing strategies that keep them away and avoid involvement with pediatric clients. Moreover, Brighton et al. [5] underlined the importance of time and space to reflect in an informal supportive process. Therefore, the emotional component of caring needs formal and systematic training to manage feelings, involving theoretical concepts of psychology, sociology, and interpersonal abilities [5,10]. Organizational systems must be close and attentive to support healthcare professionals’ mental and physical health and promote their well-being through a workplace culture that recognizes emotional experiences and supports emotion management [5,45]. The emotional labor provided a trajectory to recognize the joy and burden for the healthcare professionals, clients, and family [45].

This scoping review has some limitations. We only included publications written in English and Portuguese. The limited number of studies examining emotional labor in pediatric palliative care may justify conducting a broad search in all languages. Also, some significant publications may be missing because of the databases used in the search.

## 5. Conclusions

The review about emotional labor in palliative pediatric contexts highlighted emotional regulation strategies to face difficult situations in caring for children with palliative needs and their parents, demanding internal efforts. Strategies of emotional labor used by healthcare professionals when caring for children and their parents in palliative care ameliorate and cultivate the relationship established between children, parents, and healthcare professionals, helping them to manage intense emotions, predominantly negative emotions such as sadness and frustration. The results demonstrated that intrapersonal, social, and organizational emotional labor strategies were diversified and associated with surface acting and deep acting. These identified strategies depend on the relational context of caring and the unique characteristics of children and parents.

The team’s support was unquestionable for developing the emotional labor performance of healthcare professionals, but individual strategies are also highlighted. Managing care and emotional labor is a vital component of pediatric palliative care, improving the quality of care. Therefore, emotional labor strategies are essential to care for children and parents, and they are based on supportive reflection and education, which promote best caring practices. This review highlights the importance of emotional labor performance for the well-being of healthcare professionals in clinical practice, but this depends on training and emotional management development.

The results emphasize that healthcare organizations can provide space and time for healthcare professionals to share and discuss their emotions and concerns resulting from care for pediatric clients and offer education programs about emotional labor.

Furthermore, this scoping review contributes to awareness of the emotional dimension of healthcare because the concept of emotional labor is recent and underdeveloped. Additionally, it provides valuable insights into the deeply emotional context of pediatric palliative care. According to our findings, emotional labor in pediatric palliative care is closely linked to the emotional competence of healthcare professionals, which is characterized by intrapersonal, social, and organizational strategies.

Further research is recommended about emotional labor in healthcare professionals who care for children with palliative needs and their parents, especially research focused on strategies that help to deal with intense emotions. More research is needed to study emotional labor in pediatric palliative care, specifically how different healthcare professionals who integrate the palliative care team deal with their own emotions resulting from care for children and their parents, and focus on various contexts, such as community or family homes.

## Figures and Tables

**Figure 1 nursrep-15-00118-f001:**
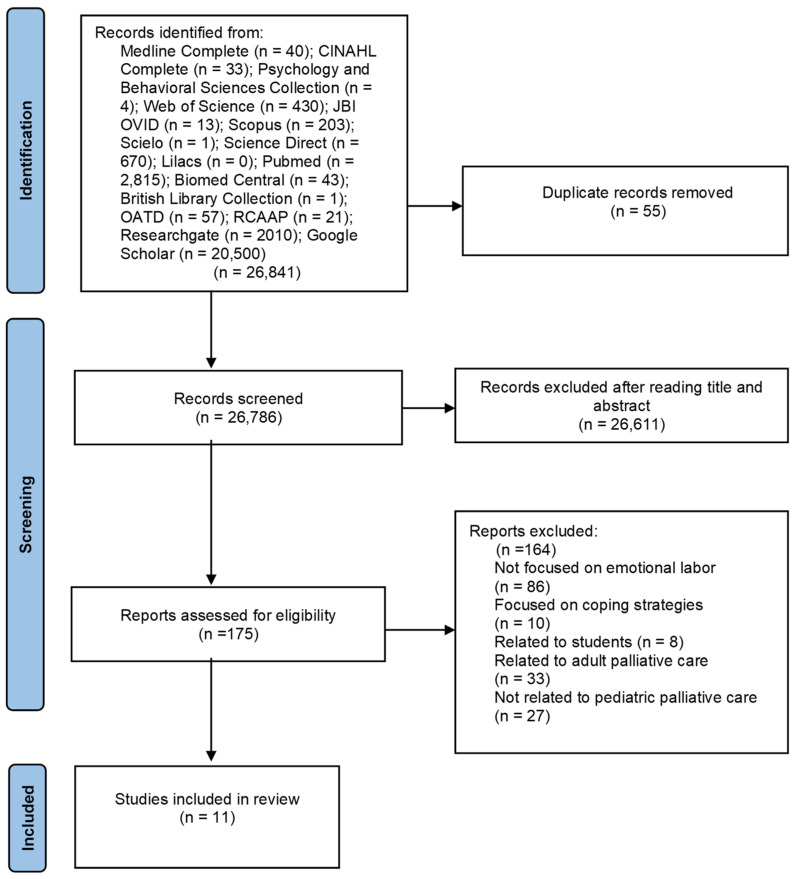
PRISMA flow diagram [22].

**Figure 2 nursrep-15-00118-f002:**
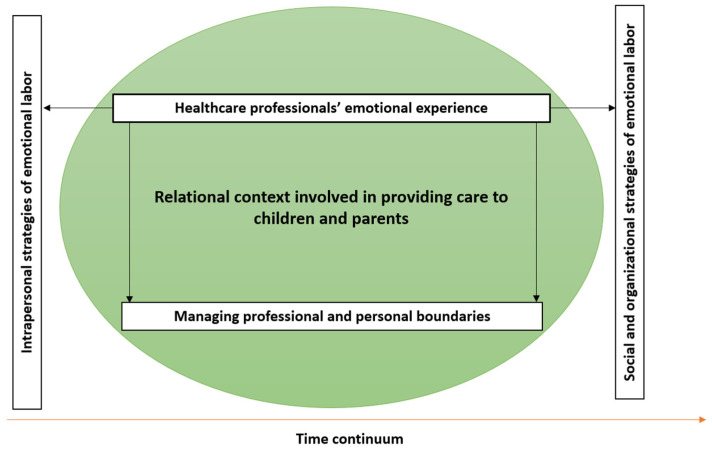
Schematic results’ representation.

**Table 1 nursrep-15-00118-t001:** The search strategy used in all databases.

Database	Search Strategy
MEDLINE Complete	(Nurses OR Physicians OR Social Workers OR Psychologists OR Patient Care Team) AND (Emotion Work OR Emotional Management OR Emotional Regulation OR Emotional Labour OR Emotional Labor) AND (Pediatric Palliative Care OR Paediatric Palliative Care OR End-of-life)
CINAHL Complete	(Nurses OR Physicians OR Social Workers OR Psychologists OR Patient Care Team) AND (Emotion Work OR Emotional Management OR Emotional Regulation OR Emotional Labour OR Emotional Labor) AND (Pediatric Palliative Care OR Paediatric Palliative Care OR End-of-life)
Psychology and Behavioral Sciences Collection	(Nurses OR Physicians OR Social Workers OR Psychologists OR Patient Care Team) AND (Emotion Work OR Emotional Management OR Emotional Regulation OR Emotional Labour OR Emotional Labor) AND (Pediatric Palliative Care OR Paediatric Palliative Care OR End-of-life)
Web of Science	(Nurses OR Physicians OR Social Workers OR Psychologists OR Patient Care Team) AND (Emotion Work OR Emotional Management OR Emotional Regulation OR Emotional Labour OR Emotional Labor) AND (Pediatric Palliative Care OR Paediatric Palliative Care OR End-of-life)
Scopus	(Nurses OR Physicians OR Social Workers OR Psychologists OR Patient Care Team) AND (Emotion Work OR Emotional Management OR Emotional Regulation OR Emotional Labour OR Emotional Labor) AND (Pediatric Palliative Care OR Paediatric Palliative Care OR End-of-life)
JBI OVID	(Nurses OR Physicians OR Social Workers OR Psychologists OR Patient Care Team) AND (Emotion Work OR Emotional Management OR Emotional Regulation OR Emotional Labour OR Emotional Labor) AND (Pediatric Palliative Care OR Paediatric Palliative Care OR End-of-life)
Scielo	(Emotional Labor OR Emotional Labour) AND (Pediatric Palliative Care OR Paediatric Palliative Care OR End-of-life)
Lilacs	(Emotional labor OR Emotional Labour) AND (Pediatric Palliative Care OR Paediatric Palliative Care)
PubMed	(Nurses OR Physicians OR Social Workers OR Psychologists OR Patient Care Team) AND (Emotion Work OR Emotional Management OR Emotional Regulation OR Emotional Labour OR Emotional Labor) AND (Pediatric Palliative Care OR Paediatric Palliative Care OR End-of-life)
Biomed Central	(Nurses OR Physicians OR Social Workers OR Psychologists OR Patient Care Team) AND (Emotion Work OR Emotional Management OR Emotional Regulation OR Emotional Labour OR Emotional Labor) AND (Pediatric Palliative Care OR Paediatric Palliative Care OR End-of-life)
Science Direct	(Patient Care Team) AND (Emotional Labor) AND (Pediatric Palliative Care)
British Library Collection	(Emotional Labor OR Emotional Labour) AND (Pediatric Palliative Care OR Paediatric Palliative Care OR End-of-life)
Open Access Theses and Dissertations (OATD)	(Emotional Labor) AND (Pediatric Palliative Care)
Open Access Scientific Repository of Portugal (RCAAP)	(Nurses OR Physicians OR Social Workers OR Psychologists OR Patient Care Team) AND (Emotion Work OR Emotional Management OR Emotional Regulation OR Emotional Labour OR Emotional Labor) AND (Pediatric Palliative Care OR Paediatric Palliative Care OR End-of-life)
ResearchGate	(Emotional Labor OR Emotional Labour) AND (Pediatric Palliative Care OR Paediatric Palliative Care)
Google Scholar	(Nurses OR Physicians OR Psychologists OR Social Workers OR Palliative Care Team) AND (Emotional Labor OR Emotional Labour) AND (Pediatric Palliative Care OR Paediatric Palliative Care)

**Table 2 nursrep-15-00118-t002:** Table of selected publications.

Author(s), Year, Country	Article Title	Methodology	Participants	Aim	Main Results	Suggestions and Limitations
Mandy J. Brimble, Sally Anstey, Jane Davies, Catherine Dunn [23] 2024 United Kingdom	“An exploration of managing emotional labour and maintaining professional integrity in children’s hospice nursing”	Narrative interpretive approach	A purposive sample of 6 children’s nurses working in a children’s hospice setting	To investigate how children’s hospice nurses manage emotional labour and professional integrity in their long-term relationships with parents	Three themes were identified: purposeful positioning (creating a psychological space between myself and work, managing empathy and emotional self-regulation); balancing personability and professionalism (I am a friendly professional, I am not their friend; managed self-disclosure); coping with and counterbalancing emotional labour (job satisfaction, positivity and fun, exceptional peer support). All themes were indicative of and built upon emotional intelligence constructs, such as self-awareness, self-regulation, empathy, social skills, and intrinsic motivation. Innate features of children’s hospice work were important for perpetuating motivation and satisfaction.	Emotional intelligence is key to managing emotional labour and professional integrity in children’s hospice nursing because can enhance relational skills. Emotional intelligence can be taught, and raising awareness is essential. Supporting well-being interventions for nurses, such as building/rebuilding resilience, is critical.
Amanda M. Gengler [24] 2023 USA	“The medicine is the easy part”: Pediatric physicians’ emotional labor in end of life	Inductive grounded theory approach	12 physicians caring for children with life-threatening medical conditions at two children’s hospitals	To illustrate the potential personal costs of emotional labor for physicians, the strategies used by physicians to perform it most effectively, and the resources that facilitated efforts to care	Findings: (1) Physician’s emotional labor: Strategic relationship-building; (2) Physicians’ emotional labor: “Laying black crepe” conceptualized themselves as most responsible for helping families through the dying process; (3) Physician’s emotional labor: Mitigating disparities in care, emotional labor as a tool that could help physicians to combat the potential for marginalized families to feel less comfortable or confident in their care; (4) “I wish there were more of me”: Balancing emotional labor and personal well-being. They learned to maintain the emotional reserves they needed for themselves and their families; (5) (Re)producing a new emotional culture in medicine. Physicians conceptualize the work as a fundamental responsibility to the children and families in their care despite believing that their formal medical training fails to prepare them for this component of their jobs.	Lower patient caseloads, longer appointment times, explicit worktime for physicians to complete notes and respond to patient messages, emotion-focused mentoring relationships between junior and senior physicians, and flexibility for physicians engaging in actions that support families at the end of life could contribute to enhancing relationships between providers and patients.
Mandy J. Brimble [25] 2023 United Kingdom	How do children’s nurses working in hospices manage emotional labour and professional integrity in long-term relationships with parents?	Thematic analysis	6 registered children’s nurses, employed at children’s hospice	To develop an understanding of how children’s hospice nurses maintain professional integrity whilst providing long-term practical, emotional, social, and spiritual care to parents and explore coping strategies used by children’s hospice nurses to manage emotional labor.	Nurses used a range of strategies to manage their relationship with parents. In terms of their emotions (purposeful positioning) and interactions (balancing personability and professionalism). Nurses revealed other children’s hospice-specific factors that helped them cope with their role (coping with and counterbalancing emotional labor). Findings were indicative of children’s hospice nurses’ using and building emotional intelligence.	Nurses used emotional intelligence to engage emotionally with parents whilst managing the level of involvement and maintaining a sense of separation.
Weina Bian, Junxiang Cheng, Yue Dong, Ying Xue, Qian Zhang, Qinghua Zhend, Rui Song, Hongwei Yang [26] 2023 China	Experience of pediatric nurses in nursing dying children—a qualitative study	Descriptive qualitative study	10 nurses from the pediatric, pediatric emergency, and neonatology departments.	To explore pediatric nurses’ challenges and effective coping strategies in caring for dying children	Themes were generated: stressors: negative emotions, such as sadness, dismay, and anxiety; helplessness; questioning rescue behavior, making nurses constantly recall the rescue process; fear of communication; and lack of workforce for night rescue. Stress consequences: compassion fatigue; burnout; changes in life attitudes. Coping strategies: self-regulation such as self-mediation, releasing their pressure by watching music, watching TV, reading books, and slowly forgetting the experience of rescuing the children. Nurses will not talk about it with their families. Some mentioned finding friends to communicate with; leadership approval and no accountability, nurses want support from a leader.	Nursing managers should pay attention to the feelings of nurses who care for dying children. Moreover, it provides some information for the development of nurses and the formulation of relevant policies.
Deb Rawlings, Megan Winsall, Huahua Yin, Kim Devery [27] 2022 Australia	“Holding back my own emotions”: Evaluation of an online education module in pediatric end-of-life care	Multi-method approach	Nurses, doctors, and allied health professionals	To evaluate online education modules and explore learners’ views on challenges faced when caring for a dying child and their family in a hospital setting	Quantitative findings: the post-evaluation ranks of learners’ perceived knowledge, skill, attitude, and confidence were statistically significantly higher compared to the pre-evaluation level. Qualitative findings: themes were organized into two categories: dealing with emotions (managing own emotions; providing adequate emotional support and comfort; witnessing the emotions and grief of families; unfairness and unnaturalness of the death of a child; identifying as a parent; feeling helpless or inadequate) and communicating effectively (answering questions, not knowing what to say; being honest about the child’s prognosis; discussing goals of care, respect, and incorporating the family’s needs and wishes)	More education is needed about pediatric end-of-life care. Clinicians refer that managing their emotions and recognize professional and personal boundaries are difficult. Debriefing with colleagues, and organizational support, such as clinical supervision, reflection, and references on self-care are important.
Beyzanur Erel and Esra Tural Büyük [28] 2021 Turkey	The effect of emotional labor levels on the attitudes of neonatal intensive care nurses towards palliative care	Correlational, cross-sectional, and descriptive design	96 nurses employed in the neonatal intensive care unit	To determine the effect of emotional labor levels of neonatal nurses on their attitudes toward palliative care	Among the NICU nurses, who participated in the study, 61.3% stated that they did not receive training in palliative care, and 84% did not find their knowledge about palliative care sufficient. A statistically significant and negative correlation was found between the surface-acting sub-dimension of the ELS and the resources sub-dimension of the NPCAS, and a statistically significant and positive correlation was found between the expression of naturally felt emotions sub-dimension of the ELS and the clinicians sub-dimension of NPCAS. Nurses suppressed their real emotions while providing palliative care and behaved superficially. Moreover, it was determined that they displayed a positive attitude towards knowing and cooperating with neonatal palliative care practices. Nurses stated that the institutional support provided to them was not sufficient.	Training on emotional resilience, communication, and emotional labor should be provided. This training should be implemented regularly and continuously, including creative drama, simulation, and role-play methods. In-service training should be provided for nurses to increase their awareness of neonatal palliative care. Guidelines should be developed.
Branislav Uhrecký, Jitka Gurňáková, Denisa Marcinechová [29] 2021 Slovak Republic	‘We Ought to be Professionals’: Strategies of Intrapersonal and Interpersonal Emotion Regulation of Emergency Medical Services Professionals in Confrontation With the Death of a Newborn in Simulated Task	Inductive and deductive approaches were combined	48 crew leaders (30 males, 18 females) from the Czech and Slovak Republic	To explore the emotion regulation strategies in a simulated task that focused on these skills	The results are structured into three sections—the affective reactions of paramedics, intrapersonal emotion regulation strategies, and interpersonal emotion regulation strategies. Intrapersonal emotional regulation includes focusing on the task (attentional narrowing, deliberate concentration, vigilance, and distancing for the scene), distancing and detachment (emotional distancing, detachment, and professionalism), and cognitive framing (positive framing, attribution to external factors, and accepting attitude). Interpersonal emotion regulation includes distraction, allowing visual contact, pretending to provide medical care, providing information, allowing space to grief, and cognitive reframing.	Regarding interpersonal emotion regulation strategies, some of them were contradictory to the optimal procedure that was proposed, such as not allowing the mother to have visual contact with her child and directing her attention elsewhere.
Paula Diogo, José Vilelas, Luiza Rodrigues, Tânia Almeida [30] 2014 Portugal	Emotional Nursing Labour in the Childcare at the End-of-Life and Their Family: A Systematic Review	Systematic Review	Nurses	To systematize scientific evidence about the emotional labor of nurses in the process of childcare at the end of life and their families	Emotional labor has focused on nurses themselves, as they are affected by the emotional responses of the clients and the need to manage these emotions in their care practice. The performance of emotional labor is characterized as a component of the care process, as a key competence in caring, as a stressful experience in confronting the suffering of the client, and as a regulation of own emotions.	This study revealed a gap in knowledge regarding the concept of nursing emotional labor in death situations, due to the reduced number of and the lack of attention to training nurses.
Roberta Cricco-Lizza [31] 2014 USA	The Need to Nurse the Nurse: Emotional Labor in Neonatal Intensive Care	Ethnographic approach	114 nurses in the neonatal intensive care unit	To inductively capture the emotional labor of nurses and explore their coping strategies during their everyday care in the neonatal intensive care unit	Nurses reported three sources of emotional demands. On a personal level, they had unique concerns originating from their private life. On a professional level, they experienced emotional challenges arising from their daily care of babies and parents. They had to deal with stressors from the demands of the organization. Nurses used individual strategies when they were outside of the hospital: exercise, rest, recreational diversions, and spiritual renewal. Nurses used multiple mechanisms to cope with troubling emotions in the workplace: talking with the sisterhood of nurses, being a super nurse, using social talk and humor to defuse intensity, taking breaks, offering flexible aid, withdrawing from emotional pain, transferring out of the unit, attending memorial services, and reframing loss to find meaning in work.	The study contributes to the development of interventions for nurses and ultimately facilitate Neonatal Intensive Care nurses’ nurturance of stressed families. These have implications for staff retention, job satisfaction, and care delivery.
Monica Restrepo, Shanna Pilgrim [32] 2011 USA	Caring for the Caregiver: Emotional Challenges of Pediatric Palliative Care Nurses	Literature Review	Pediatric palliative care nurses (PPC nurses)	This paper looks at how PPC nurses cope with caregiver emotions within the conceptual framework of emotional labor and emotional intelligence	Emotional labor allows PPC nurses to put on an appropriate face through surface acting when dealing with difficult and traumatic situations when caring for dying children and their families; the frequency of interaction and intensity of emotions plays a unique role with PPC nurses; Emotional dissonance resonates strongly in PPC nursing; Emotional intelligence allows nurses to acknowledge their emotions, rather than suppress their emotions; the use of emotional labor is an explicit tool for self-protection and how emotional intelligence could be useful in ameliorating the backside of those stored emotions.	Pediatric palliative care nursing programs benefit from implementing training and education components that include emotional labor and emotional intelligence.
Eryl Zac Maunder [33] 2008 United Kingdom	Emotion management in children’s palliative care nursing	Literature Review	Nurses	To explore emotional labor involved for nurses providing palliative care for children/young people living with life-limiting illnesses/conditions and their families	It highlights the challenges nurses face in managing their emotions when caring for children/young people and their families and explores strategies to enable nurses to cope with this aspect of their role without compromising their well-being. It suggests that emotional labor within nursing goes largely unrecorded and remains undervalued by managers and healthcare services.	When work conditions to support emotional labor are good, and the nursing team is encouraged to work to their areas of expertise, the nurses feel better supported.

**Table 3 nursrep-15-00118-t003:** Characteristics of the publications.

Characteristics	Number of Publications
Year of Publication	Number of Publications
2024	1
2023	3
2022	1
2021	2
2020	0
2019	0
2018	0
2017	0
2016	0
2015	0
2014	2
2013	0
2012	0
2011	1
2010	0
2009	0
2008	1
Country	Number of Publications
United States of America	3
United Kingdom	3
Portugal	1
Turkey	1
Australia	1
Slovak Republic	1
China	1
Type of Publications	Number of Publications
Qualitative Studies	6
Descriptive qualitative study	
Ethnographic approach	
Grounded theory	
Thematic analysis	
Inductive and deductive approach	
Narrative interpretative approach	
Quantitative Study	1
Mixed Methods Study	1
Systematic Literature Review	1
Literature Review	2
Type of Healthcare Professionals *	Number of Publications
Nurses	9
Physicians	3
Other Healthcare Professionals	1
Emergency Medical Services Professionals (Paramedic/Paramedic Driver)	1
Context of Pediatric Palliative Care *	Number of Publications
Neonatal Intensive Care Unit	3
Pediatric Department	2
Children’s Hospice	2
Emergency Department	1
Simulated Task	1
Without Reference	3

* More than one category can be applied to one publication.

**Table 4 nursrep-15-00118-t004:** Healthcare professionals’ emotional experiences.

Emotions Experienced	References
Burnout, grief, and stress	[32]
Sadness, dismay, and anxiety	[26,32]
Feeling of helplessness	[26]
Feeling of powerlessness	[27]
Overwhelmed	[29]
Frustration and anger	[32]
Suffering	[27]

**Table 5 nursrep-15-00118-t005:** Relational context involved.

Relationship’s Characteristics	References
Close and supportive	[30,31]
Intentional and intimate	[23,24,25]
Healthcare professional like a family member	[33]
Planned and strategic	[24]
Confidence	[23,24]
Support Provided	References
Meeting the child’s and the parents’ needs	[27,33]
Providing emotional support and comfort	[24,27]
Venting concerns	[31]
Offering empathy and compassion	[27]
Providing a mental break and distraction	[31]
Information, guidance, emotional, instrumental, and financial support	[30]
Open and effective communication	[27,30]

**Table 6 nursrep-15-00118-t006:** Managing professional and personal boundaries.

Characteristics	References
Maintaining personal and professional boundaries	[27,33]
Maintaining professional integrity	[25]
Balancing personability and professionalism	[23,25]
Balancing emotional labor and personal well-being	[24]
Tension between commitment to the client’s support and personal emotional care	[24]

**Table 7 nursrep-15-00118-t007:** Intrapersonal strategies of emotional labor.

Strategies Used to Deal with Own Emotions Arising from the Care of Children and Parents	References
Avoiding becoming overwhelmed	[27,29]
Hiding personal sadness	[27]
Suppressing real emotions and behaving superficially	[28]
Putting an appropriate face	[32]
Keeping emotional distance	[30]
Distancing from affective responses	[29]
Building an emotional barrier	[33]
Indifference	[30]
Task orientation and super-efficient attention to the child	[29,31,33]
Compartmentalization	[24,27,31]
Cognitive reinterpretation	[28]
Finding a positive meaning	[30,31]
Making memories	[23]
Attending memorial services	[24,31]
Service transference	[31]
Self-care practices	References
Exercise, rest, reading a book, listening to music, watching TV	[26,31]
Taking breaks	[31]
Seeking spiritual strategies	References
Self-meditation, religion, faith, spiritual restoration, spiritual renewal	[26,30,31]

**Table 8 nursrep-15-00118-t008:** Social and organizational strategies of emotional labor.

Social and Organizational Strategies	References
Support from friends or colleagues	[26]
Support from the team	[23,30,33]
Support from managers/leaders	[26,31]
Use of humor	[31,33]
Professional development strategies	References
Education and training programs	[24,28,31]
Debriefing sessions	[31]

## Data Availability

No new data were created or analyzed in this study. Data sharing is not applicable to this article.

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
