# Peer review of "Emotional Labor in Pediatric Palliative Care: A Scoping Review"

_nursrep, 2025, doi:10.3390/nursrep15040118_

Round 1

Reviewer 1 Report

Comments and Suggestions for Authors

The manuscript is well-written and makes a valuable contribution to pediatric palliative nursing. It adheres to the JBI and PRISMA recommendations, reflecting the rigour of the work.

Below are several suggestions to further enhance the quality and impact of the manuscript:

Consistency in Terminology: In the summary section, it is noted that a thematic analysis was performed to synthesize the findings (see line 18). However, in line 22, content analysis is mentioned, which describes five themes. It is recommended that the authors maintain consistency in terminology throughout the manuscript to avoid any confusion.

Introduction Section: In line 65, it is suggested that the author (Hochschild) be included when referring to deep and surface acting. Similarly, in line 68, the same author should be cited when mentioning the 'feeling rules.' Given that Hochschild's framework is being utilized, it may also be worth considering the inclusion of a discussion on how emotional labour in the work context is often linked to an exchange of money. This could be a valuable addition to future research and may prompt readers to reflect on the manuscript's claim that emotional labour is "often perceived as undervalued."

Materials and Methods Section: In line 152 of Table 1, it is recommended that each database include the actual search strategy used, as the same strategy is listed for all of them. Providing specific details for each database would increase the rigour and replicability of the search process.

Flow Diagrams: The manuscript includes two flow diagrams. It is suggested that the second diagram, which includes the articles identified by the database and the total number of studies included (11), should be retained, as indicated in the abstract. This will provide clarity and align the diagram with the content of the manuscript.

In the Data Analysis and Presentation section, it is recommended that the authors address how potential limitations of the synthesis process were managed, particularly the variability among the studies included.

In line 184, content analysis is mentioned. It is suggested that the authors review the earlier comment about maintaining consistency in terminology throughout the manuscript.

For lines 222, 227, 278, 306, and 314, it is advised that the authors include appropriate references to support the claims or information presented in these sections.

In line 464, it is recommended to review the citation to ensure it is correctly formatted and accurately reflects the referenced work.

In the Discussion section, it would be helpful for the authors to explain the rationale behind the schematic representation of the results and why the categories were mapped visually in the particular way they were. This clarification would improve the reader's understanding of the data's interpretation. Additionally, it is suggested that a brief statement regarding the quality of the studies in the synthesis be included, as this would provide valuable context for the findings.

Author Response

Reviewer 1

The authors thank the reviewer for the suggestions and analysis that were accepted, hoping these changes contribute to improve the quality of the manuscript.

Comment 1:

Consistency in Terminology: In the summary section, it is noted that a thematic analysis was performed to synthesize the findings (see line 18). However, in line 22, content analysis is mentioned, which describes five themes. It is recommended that the authors maintain consistency in terminology throughout the manuscript to avoid any confusion.

Response 1:

We thank you for pointing this inconsistency, and we have changed to “content analysis”.

Comment 2:

Introduction Section: In line 65, it is suggested that the author (Hochschild) be included when referring to deep and surface acting. Similarly, in line 68, the same author should be cited when mentioning the 'feeling rules.' Given that Hochschild's framework is being utilized, it may also be worth considering the inclusion of a discussion on how emotional labour in the work context is often linked to an exchange of money. This could be a valuable addition to future research and may prompt readers to reflect on the manuscript's claim that emotional labour is "often perceived as undervalued."

Response 2:

We agree with this comment; changes were made and highlighted in yellow in the manuscript. 

Comment 3:

Materials and Methods Section: In line 152 of Table 1, it is recommended that each database include the actual search strategy used, as the same strategy is listed for all of them. Providing specific details for each database would increase the rigour and replicability of the search process.

Response 3:

Thank you for pointing this out. In line 152, table 1, we presented the specific search strategy used in each database. These search strategies were used in the first computerized literature search and in the updated search done on February 17, 2025.

Comment 4:

Flow Diagrams: The manuscript includes two flow diagrams. It is suggested that the second diagram, which includes the articles identified by the database and the total number of studies included (11), should be retained, as indicated in the abstract. This will provide clarity and align the diagram with the content of the manuscript.

Response 4:

In response to your suggestion, we recognize the lapse. We have now included only one PRISMA flow diagram, which contains the articles identified in each database and the total number of studies included (11).

Comment 5:

In the Data Analysis and Presentation section, it is recommended that the authors address how potential limitations of the synthesis process were managed, particularly the variability among the studies included.

Response 5:

We agree with this comment; changes were made and highlighted in yellow in the manuscript. 

Comment 6:

In line 184, content analysis is mentioned. It is suggested that the authors review the earlier comment about maintaining consistency in terminology throughout the manuscript.

Response 6:

Thank you for pointing this out. We modified the manuscript according to the reviewer's comment. We decided to change to “content analysis”.

Comment 7:

For lines 222, 227, 278, 306, and 314, it is advised that the authors include appropriate references to support the claims or information presented in these sections.

Response 7:

We agree with these comments; changes were made and highlighted in yellow in the manuscript.

Comment 8:

In line 464, it is recommended to review the citation to ensure it is correctly formatted and accurately reflects the referenced work.

Response 8:

We agree with this comment; changes were made and highlighted in yellow in the manuscript.

Comment 9:

In the Discussion section, it would be helpful for the authors to explain the rationale behind the schematic representation of the results and why the categories were mapped visually in the particular way they were. This clarification would improve the reader's understanding of the data's interpretation. Additionally, it is suggested that a brief statement regarding the quality of the studies in the synthesis be included, as this would provide valuable context for the findings.

Response 9:

Thank you for pointing this out. We included two sentences explaining the schematic representation of the results and the methodological quality of studies identified in the scoping review.

Reviewer 2 Report

Comments and Suggestions for Authors

thank you to give me an Opportunity to review this paper , my comments for improvement :

This introduction effectively sets the stage for the importance of understanding emotional labor in pediatric palliative care. It clearly emphasizes the emotional challenges healthcare professionals face while providing care, but could benefit from a more concise presentation of the background research to avoid overwhelming the reader.

The article provides an insightful overview of the role emotional labor plays in pediatric palliative care, highlighting the gaps in current research. It sets a strong foundation for the scoping review, though the dense theoretical framework could be simplified for better accessibility to a broader audience.

The methodology is well-structured and clearly follows established guidelines, such as the PRISMA checklist. However, providing a brief explanation of the rationale behind choosing specific databases would enhance the transparency of the search strategy."

The eligibility criteria are thorough and well-defined, making the scope of the review clear. It would be helpful to include more details on how data extraction and analysis were performed, as this could provide a better understanding of the review's robustness

Author Response

Reviewer 2

The authors thank the reviewer for the suggestions and analysis that were accepted, hoping these changes contribute to improve the manuscript's quality.

Comment 1:

This introduction effectively sets the stage for the importance of understanding emotional labor in pediatric palliative care. It clearly emphasizes the emotional challenges healthcare professionals face while providing care, but could benefit from a more concise presentation of the background research to avoid overwhelming the reader.

Response 1:

We agree with this comment; changes were made in the manuscript.

Comment 2:

The article provides an insightful overview of the role emotional labor plays in pediatric palliative care, highlighting the gaps in current research. It sets a strong foundation for the scoping review, though the dense theoretical framework could be simplified for better accessibility to a broader audience.

 Response 2:

We agree with this comment; changes were made in the manuscript.

Comment 3:

The methodology is well-structured and clearly follows established guidelines, such as the PRISMA checklist. However, providing a brief explanation of the rationale behind choosing specific databases would enhance the transparency of the search strategy.

Response 3:

We agree with this comment; changes were made and highlighted in yellow in the manuscript.

Comment 4:

The eligibility criteria are thorough and well-defined, making the scope of the review clear. It would be helpful to include more details on how data extraction and analysis were performed, as this could provide a better understanding of the review's robustness.

Response 4:

Thank you for pointing this out. We included a sentence detailing how data extraction and analysis were performed.

Reviewer 3 Report

Comments and Suggestions for Authors

This scoping review covers a highly relevant topic and provides valuable insights. The research question is well-defined and justified, and the study makes a meaningful contribution to the field.

  • You referenced a 2020 scoping review that identified 29 publications and highlighted the relevance of research on emotional labor in pediatric nursing. This current scoping review expands the scope beyond nurses to include other healthcare professionals, which is a significant and interesting contribution. However, why was "no time limit" set in the eligibility criteria? Could you clarify the rationale behind this decision?
  • You mentioned excluding studies involving students but retaining those where students were also included. Were you able to effectively exclude them when reporting the results?
  • Additionally, you screened the titles and abstracts of 197,535 studies. Have you considered that this might indicate an issue with the excessive sensitivity of the search strings? It might be worth revising the search strategy to improve specificity. Furthermore, I would recommend reconsidering the inclusion of the "Open Access Theses and Dissertations" (OATD) database, as its scientific value is debatable.

  • The results are well-organized, but their presentation could be enhanced through clearer tables, figures, or conceptual maps to improve readability and comprehension.
    The conceptual map you included is unclear and difficult to interpret. A more structured approach, such as using a table categorizing thematic areas, might improve clarity. You stated: "This scoping review aims to identify and map the scientific production," but the mapping appears inconsistent and lacks a clear methodological approach.

  • It would have been valuable to further elaborate on the conceptual analyses by categorizing the findings based on the different types of healthcare professionals included in the study.

  • Although a scoping review does not assess the quality of the included studies, it would still be useful to provide a general discussion on the risk of bias and the limitations of the analyzed literature. While you correctly acknowledge the language limitation, highlighting additional research gaps would strengthen the review.

  • The discussion would benefit from a more in-depth exploration of how the findings can be applied in practice and future research. Providing concrete implications for different healthcare professionals could further strengthen the impact of the study.

Overall, this scoping review makes a valuable contribution to the field. Addressing these points would improve its clarity, methodological rigor, and practical applicability.

Author Response

Reviewer 3

The authors thank the reviewer for the suggestions which were accepted, hoping they contribute to improving the manuscript’s quality.

Comment 1:

You referenced a 2020 scoping review that identified 29 publications and highlighted the relevance of research on emotional labor in pediatric nursing. This current scoping review expands the scope beyond nurses to include other healthcare professionals, which is a significant and interesting contribution. However, why was "no time limit" set in the eligibility criteria? Could you clarify the rationale behind this decision?

Response 1:

Thank you for pointing this out. The article “Emotional labor of nursing: a scoping review on pediatric care contexts” aimed to identify and systematize available publications about emotional labor in diverse pediatric nursing care contexts, that explain and sustain emotional labor in pediatric nursing care, demonstrated that the most studied pediatric nursing contexts were pediatric inpatient service, pediatric palliative care and neonatal intensive care. In this new scoping review, we explore emotional labor performed by healthcare professionals in the pediatric palliative care context, so we decided “no time limit”. This selection was related to a pediatric palliative care team as a population, not only nurses, who cared for children with palliative needs and their parents.

Comment 2:

You mentioned excluding studies involving students but retaining those where students were also included. Were you able to effectively exclude them when reporting the results?

Response 2:

We agree with this comment. We confirm that we excluded all publications referring to healthcare students in this scoping review. In data extraction and data selection, we effectively excluded students.

Comment 3:

Additionally, you screened the titles and abstracts of 197,535 studies. Have you considered that this might indicate an issue with the excessive sensitivity of the search strings? It might be worth revising the search strategy to improve specificity. Furthermore, I would recommend reconsidering the inclusion of the "Open Access Theses and Dissertations" (OATD) database, as its scientific value is debatable.

 Response 3:

Thank you for pointing this out. We revised the search strategy in each database to improve specificity. Considering this is a scoping review aiming to determine the coverage of literature on the topic of emotional labor in the context of pediatric palliative care, we decided to include master's dissertations and doctoral theses. To capture these studies, we included databases, including these publications, such as the  Open Access Theses and Dissertations (OATD). OATD is not a scientific database in the traditional sense, but it is a valuable source for grey literature. So, conducting a scoping review, it can be included to capture unpublished research and broader perspectives. However, it was supplemented with established scientific databases.

Comment 4:

The results are well-organized, but their presentation could be enhanced through clearer tables, figures, or conceptual maps to improve readability and comprehension.

Response 4:

We agree with this comment and we decided to include tables on each theme to improve comprehension of the results.

Comment 5:

The conceptual map you included is unclear and difficult to interpret. A more structured approach, such as using a table categorizing thematic areas, might improve clarity. You stated: "This scoping review aims to identify and map the scientific production," but the mapping appears inconsistent and lacks a clear methodological approach.

 Response 5:

Thank you for pointing this out. We decided to do a more structured approach of the schematic representation and include a sentence explaining that.

Comment 6:

It would have been valuable to further elaborate on the conceptual analyses by categorizing the findings based on the different types of healthcare professionals included in the study.

Response 6:

Thank you for pointing this out. We include tables categorizing the themes of the results in the manuscript. Nurses are the most healthcare professionals involved in the included studies.

Comment 7:

Although a scoping review does not assess the quality of the included studies, it would still be useful to provide a general discussion on the risk of bias and the limitations of the analyzed literature. While you correctly acknowledge the language limitation, highlighting additional research gaps would strengthen the review.

Response 7:

We agree with the comment and include a sentence in the manuscript.

Comment 8:

The discussion would benefit from a more in-depth exploration of how the findings can be applied in practice and future research. Providing concrete implications for different healthcare professionals could further strengthen the impact of the study.

 Response 8:

Thank you for pointing this out. In conclusion, we include sentences explaining how findings can be applied in practice and future research.

Round 2

Reviewer 3 Report

Comments and Suggestions for Authors

Thanks for your answers, I'm sure that your work, which was already well done, is now clearer and more complete. Congratulations.